# Magnetic moment of electrons in systems with spin-orbit coupling

I. A. Ado[1]⋆, M. Titov[1], Rembert A. Duine[2,3] and Arne Brataas[4]

**1** Radboud University, Institute for Molecules and Materials,
Heyendaalseweg 135, 6525 AJ Nijmegen, The Netherlands
**2** Institute for Theoretical Physics, Utrecht University,
Princetonplein 5, 3584 CC Utrecht, The Netherlands
**3** Department of Applied Physics, Eindhoven University of Technology,
P.O. Box 513, 5600 MB Eindhoven, The Netherlands
**4** Center for Quantum Spintronics, Department of Physics, Norwegian University of Science and
Technology, Høgskoleringen 5, 7491 Trondheim, Norway

⋆ iv.a.ado@gmail.com

April 10, 2025

## Abstract

**Magnetic effects originating from spin-orbit coupling (SOC) have been attracting major attention. However, SOC contributions to the electron magnetic moment operator are conventionally disregarded. In this work, we analyze relativistic contributions to the latter operator, including those of the SOC-type: in vacuum, for the semiconductor 8 band Kane model, and for an arbitrary system with two spectral branches. In this endeavor, we introduce a notion of relativistic corrections to the operation $\partial/\partial B$, where $B$ is an external magnetic field. We highlight the difference between the magnetic moment and $-\partial H/\partial B$, where $H$ is the system Hamiltonian. We suggest to call this difference the abnormal magnetic moment. We demonstrate that the conventional splitting of the total magnetic moment into the spin and orbital parts becomes ambiguous when relativistic corrections are taken into account. The latter also jeopardize the "modern theory of orbital magnetization" in its standard formulation. We derive a linear response Kubo formula for the kinetic magnetoelectric effect projected to individual branches of a two branch system. This allows us, in particular, to identify a source of this effect that stems from noncommutation of the position and $\partial/\partial B$ operators' components. This is an analog of the contribution to the Hall conductivity from noncommuting components of the position operator. We also report several additional observations related to the electron magnetic moment operator in systems with SOC and other relativistic corrections.**

# 1   Introduction

The notion of magnetic moment of electron is widely used in the analysis of multiple phenomena in spintronics and magnetism. Some of those are relativistic effects, i.e. they vanish in the limit $c \to \infty$, where $c$ is the speed of light. The spin Edelstein effect (SEE) [1], the spin-orbit torques (SOT) [2], and the spin Hall effect (SHE) [3] are prominent examples of such effects. Diagonal components of the orbital magnetoelectric response tensor are also reported to be relativistic [4].

In vacuum, description of relativistic effects is obtained by considering the full 4-component structure of Dirac spinors, without performing projections to the electronic branch. In solids, the complete description requires, in principal, considering the full band structure, again without projecting to individual bands. Notably, sometimes calculations for solids are performed in the fully relativistic Dirac picture [5–7].

In some cases, e.g. for theoretical modeling, it is convenient to decouple [8–12] different bands from each other and consider contributions to the overall effect from the most relevant of them separately (often from just a single one). As a result of the decoupling, individual bands acquire spin-orbit coupling (SOC) and other relativistic corrections [9–13]. In this "decoupled" picture, SOC becomes a necessary ingredient for SEE, SOT, SHE, and other relativistic effects.

For microscopic analysis of the latter within this picture, it is thus imperative to compute SOC corrections to quantum mechanical operators projected to the bands. This is usually consistently done for Hamiltonians as they are the actual subjects of the decoupling procedure and this decoupling is basically a process of their diagonalization. SOC corrections to other operators, however, are often computed incorrectly [11, 12]. Astonishingly, for the electron magnetic moment operator, they have not been computed at all (to the best of our knowledge). As a result, important contributions to SEE, SOT, SHE, and other effects have been neglected.

Another difficulty with band projecting for systems with SOC occurs in linear response theories. Namely, such theories should consistently take into account not only the intraband matrix elements of observables and perturbations, but the interband ones as well. Recently we demonstrated how this is achieved for the electric dc conductivity [11]. For effects related to magnetic moments, a similar treatment is still missing.

In this work, we compute relativistic corrections to the electron magnetic moment operator in vacuum and in the conduction band of the Kane model [14]. We show that, because of these

corrections, the latter operator is not equal to $-\partial H/\partial \boldsymbol{B}$, where $H$ is the Hamiltonian of the system. This is analogous to the difference between the velocity operator and $\partial H/\partial \boldsymbol{p}$, where $\boldsymbol{p}$ is the canonical momentum. We also derive a general Kubo formula for the dc kinetic magnetoelectric effect [15, 16] in the presence of SOC, as an example of a correct linear response theory that involves magnetic moments and takes SOC into account. It allows us to identify a particular origin of this effect in the form of certain noncommuting operators. This is similar to contributions to the Hall effect originating in the noncommutation of the position operators [11]. We also report several other observations related to magnetic moments of electrons.

## 2 Operator of magnetic moment in vacuum

Let us consider the relativistic Dirac (D) Hamiltonian

$$H^{(D)} = \beta m c^2 + c \boldsymbol{\alpha} \boldsymbol{\pi} + V, \quad \boldsymbol{\pi} = \boldsymbol{p} - (e/c)\boldsymbol{A}, \quad \boldsymbol{\alpha} = \begin{pmatrix} 0 & \boldsymbol{\sigma} \\ \boldsymbol{\sigma} & 0 \end{pmatrix}, \quad \beta = \begin{pmatrix} 1 & 0 \\ 0 & -1 \end{pmatrix}, \quad (1)$$

where $\boldsymbol{p} = -i\hbar\boldsymbol{\nabla}$ and $\boldsymbol{\sigma}$ denotes a vector composed of Pauli matrices. By choosing the symmetric gauge $\boldsymbol{A} = [\boldsymbol{B} \times \boldsymbol{r}]/2$ for the constant magnetic field $\boldsymbol{B}$, we obtain the following definition for the total magnetic moment operator:

$$\boldsymbol{\mu}^{(D)} = -\frac{\partial H^{(D)}}{\partial \boldsymbol{B}} = \frac{e}{2}[\boldsymbol{r} \times \boldsymbol{\alpha}]. \quad (2)$$

Our goal is to project $\boldsymbol{\mu}^{(D)}$ to the electronic branch decoupled from positrons and to compute the leading order relativistic corrections to the conventional expression used for electrons, i.e.

$$\boldsymbol{\mu}_{\text{conv}} = -\mu_B \left( \boldsymbol{\sigma} + \frac{1}{\hbar}[\boldsymbol{r} \times \boldsymbol{p}] \right), \quad (3)$$

where $\mu_B = -e\hbar/2mc$ is the Bohr magneton.

### 2.1 From Dirac picture to Pauli picture

First, we need to perform the branch decoupling. For this, we apply the standard approach described in Refs. [8–11] and use a unitary transformation of the form

$$U = e^{iW}, \quad W = \begin{pmatrix} 0 & w \\ w^\dagger & 0 \end{pmatrix}, \quad (4)$$

to diagonalize the Dirac Hamiltonian, $H_U^{(D)} = U^\dagger H^{(D)} U$. The result is

$$H_U^{(D)} = \begin{pmatrix} H^{(\text{el})} & 0 \\ 0 & H^{(\text{p})} \end{pmatrix}, \quad (5)$$

where $H^{(\text{el})}$ and $H^{(\text{p})}$ are the Hamiltonians of electrons (el) and positrons (p), respectively. $H^{(\text{el})}$ and $H^{(\text{p})}$ are decoupled from each other since the off-diagonal elements in Eq. (5) got nullified. In the presence of external fields, $U$ is computed perturbatively as an expansion in powers of $1/c^2$ [8,11]. The leading and subleading order contributions to $U$ are determined by [11]

$$w = \frac{i\boldsymbol{\sigma}\boldsymbol{\pi}}{2mc} + \frac{\hbar}{4m^2c^3}(\boldsymbol{\sigma}\boldsymbol{\nabla})V - \frac{i(\boldsymbol{\sigma}\boldsymbol{\pi})^3}{6m^3c^3}. \quad (6)$$

To the order $1/c^2$, this translates into the following expression for the electron Hamiltonian [11][1]

$$H^{(\text{el})} = \frac{\pi^2}{2m} + \frac{\hbar}{4m^2c^2}[\nabla V \times \pi]\sigma + \frac{\hbar^2}{8m^2c^2}\nabla^2 V + V + \mu_{\text{B}}\sigma B - \mu_{\text{B}}\frac{\pi^2\sigma B}{2m^2c^2} - \frac{\pi^4}{8m^3c^2}, \quad (7)$$

with the energy origin shifted by $mc^2$ and the term $\propto \mu_{\text{B}}^2 B^2$ disregarded (as the magnetic field is assumed to be small). Below, we additionally disregard terms nonlinear with respect to $B$ that enter Eq. (7) as contributions to powers of $\pi$. Let us also note that the vector potential in $H^{(\text{el})}$ is allowed to comprise contributions other than those that define the magnetic field $B$.

In contrast with the transformed Hamiltonian, the transformed magnetic moment operator $\mu_U^{(\text{D})} = U^\dagger \mu^{(\text{D})} U$ is not diagonal and has the following structure:

$$\mu_U^{(\text{D})} = \begin{pmatrix} \mu^{(\text{el})} & \delta\mu \\ \delta\mu^\dagger & \mu^{(\text{p})} \end{pmatrix}. \quad (8)$$

We are interested in computing the magnetic moment of electrons $\mu^{(\text{el})}$. One may expect that this can be done by differentiating the Hamiltonian of electrons $H^{(\text{el})}$ with respect to $B$. However, this is not the case because $H^{(\text{el})}$ depends on the transformation $U$, which itself is a function of $B$. Therefore, the operation $\partial/\partial B$ is not invariant under this transformation.

Clearly, $\mu^{(\text{el})}$ can be computed by directly applying $U$ and $U^\dagger$ to $\mu^{(\text{D})}$, but this is demanding in terms of algebra. We can take an alternative route by introducing a certain auxiliary operator that will also prove useful for the formulation of the linear response theory in Sec. 4.

## 2.2 Derivative with respect to magnetic field as an operator

Let us rewrite the definition of Eq. (2) in the form

$$\mu^{(\text{D})} = [\mathfrak{B}^{(\text{D})}, H^{(\text{D})}], \quad \mathfrak{B}^{(\text{D})} = \begin{pmatrix} \mathfrak{B} & 0 \\ 0 & \mathfrak{B} \end{pmatrix}, \quad \mathfrak{B} = -\frac{\partial}{\partial B}, \quad (9)$$

that resembles the relation between the position and velocity operators. Unitary transformations preserve commutators, and therefore $\mu_U^{(\text{D})} = [\mathfrak{B}_U^{(\text{D})}, H_U^{(\text{D})}]$, where[2]

$$\mathfrak{B}_U^{(\text{D})} = U^\dagger \mathfrak{B}^{(\text{D})} U, \quad \mathfrak{B}_U^{(\text{D})} = \begin{pmatrix} \mathfrak{B}^{(\text{el})} & \delta\mathfrak{B} \\ \widetilde{\delta}\mathfrak{B} & \mathfrak{B}^{(\text{p})} \end{pmatrix}. \quad (10)$$

Moreover, since $H_U^{(\text{D})}$ is diagonal, we deduce from Eqs. (5), (8) that $\mu^{(\text{el})} = [\mathfrak{B}^{(\text{el})}, H^{(\text{el})}]$. Hence, the magnetic moment operator projection to the electronic branch $\mu^{(\text{el})}$ is fully determined by the projections $\mathfrak{B}^{(\text{el})}$ and $H^{(\text{el})}$. This is analogous to the relation between the projections of the velocity and position operators [11], $v^{(\text{el})} = [r^{(\text{el})}, H^{(\text{el})}]/i\hbar$.

The leading and subleading order contributions to $\mathfrak{B}^{(\text{el})}$ are provided by[3]

$$\mathfrak{B}^{(\text{el})} = \mathfrak{B} - \frac{1}{2}\left([\mathfrak{B}, w]w^\dagger - w[\mathfrak{B}, w^\dagger]\right) = \mathfrak{B} + \frac{1}{8m^2c^2}\left[\frac{\partial(\sigma\pi)}{\partial B}(\sigma\pi) - (\sigma\pi)\frac{\partial(\sigma\pi)}{\partial B}\right]. \quad (11)$$

Employing the symmetric gauge for $B$, we obtain

$$\mathfrak{B}^{(\text{el})} = \mathfrak{B} + \frac{i\mu_{\text{B}}}{8\hbar mc^2}\left([r \times [\pi \times \sigma]] - [[\pi \times \sigma] \times r]\right), \quad (12)$$

where terms inside the parenthesis should be understood as SOC corrections to the operation $\mathfrak{B} = -\partial/\partial B$ projected to the electronic branch decoupled from positrons.

---

[1]The 6th term in Eq. (7) was neglected in Ref. [11], where $\pi^4$ was used instead of $(\sigma\pi)^4$ in the 5th term of Eq. (17).

[2]We note that $\mathfrak{B}^{(\text{D})}$ is non-Hermitian and, in particular, $\widetilde{\delta}\mathfrak{B} \neq \delta\mathfrak{B}^\dagger$.

[3]See, for example, Eq. (28) of Ref. [12].

## 2.3   Pauli picture

We can now compute the electron magnetic moment operator $\boldsymbol{\mu}^{(\mathrm{el})} = [\mathfrak{B}^{(\mathrm{el})}, H^{(\mathrm{el})}]$. For this, we need to commute the right hand sides of Eqs. (12) and (7). Commutation with $\mathfrak{B}$ corresponds to differentiating Eq. (7) with respect to $-\boldsymbol{B}$. This gives the "naive" magnetic moment

$$\boldsymbol{\mu}^{(\mathrm{el})}_{\mathrm{naive}} = -\frac{\partial H^{(\mathrm{el})}}{\partial \boldsymbol{B}} = -\mu_{\mathrm{B}}\left(\boldsymbol{\sigma} + \frac{1}{\hbar}\left[\boldsymbol{r} \times m\boldsymbol{v}^{(\mathrm{el})}_{\mathrm{naive}}\right] - \frac{\pi^2 \boldsymbol{\sigma}}{2m^2 c^2}\right), \tag{13}$$

where we used the "naive" velocity operator computed to the order $1/c^2$,

$$\boldsymbol{v}^{(\mathrm{el})}_{\mathrm{naive}} = \frac{\partial H^{(\mathrm{el})}}{\partial \boldsymbol{p}} = \frac{\boldsymbol{\pi}}{m} - \frac{\hbar}{4m^2 c^2}[\boldsymbol{\nabla}V \times \boldsymbol{\sigma}] - \frac{\pi^2 \boldsymbol{\pi}}{2m^3 c^2}, \tag{14}$$

This is not yet the final result, as one should also commute the SOC terms of Eq. (12) with $\pi^2/2m$ and $V$ in Eq. (7). Making use of the relations $[\boldsymbol{\pi}, V] = -i\hbar\boldsymbol{\nabla}V$ and $[\boldsymbol{r}, \pi^2] = 2i\hbar\boldsymbol{\pi}$, we find to the order $1/c^2$:

$$\boldsymbol{\mu}^{(\mathrm{el})} = -\mu_{\mathrm{B}}\left(\boldsymbol{\sigma} + \frac{1}{\hbar}\left[\boldsymbol{r} \times m\boldsymbol{v}^{(\mathrm{el})}\right] - \frac{3\pi^2 \boldsymbol{\sigma} - \boldsymbol{\pi}(\boldsymbol{\pi}\boldsymbol{\sigma})}{4m^2 c^2}\right), \tag{15}$$

where the correct velocity operator

$$\boldsymbol{v}^{(\mathrm{el})} = \frac{\boldsymbol{\pi}}{m} - \frac{\hbar}{2m^2 c^2}[\boldsymbol{\nabla}V \times \boldsymbol{\sigma}] - \frac{\pi^2 \boldsymbol{\pi}}{2m^3 c^2} \tag{16}$$

differs from $\partial H^{(\mathrm{el})}/\partial \boldsymbol{p}$ [11].

As can be seen, the leading order relativistic corrections to $\boldsymbol{\mu}_{\mathrm{conv}} = -\mu_{\mathrm{B}}(\boldsymbol{\sigma} + [\boldsymbol{r} \times \boldsymbol{p}]/\hbar)$ are provided by:

- the fraction inside the parenthesis in Eq. (15),

- the difference between $\boldsymbol{p}$ and $m\boldsymbol{v}^{(\mathrm{el})}$.

It is imperative to take these corrections into account for numerical analysis of relativistic contributions to the magnetoelectric [4] and other effects.

## 2.4   Modern theory of orbital magnetization

In solids, equilibrium orbital magnetization[4] is often computed using the "modern theory of orbital magnetization" [17–21] that averages the observable $\mu_{\mathrm{orb}} = -(\mu_{\mathrm{B}}/\hbar)[\boldsymbol{r} \times m\boldsymbol{v}]$ over the Bloch states and system volume. This theory is derived in the absence of relativistic corrections (i.e. in a fully nonrelativistic picture) and assumes that the velocity and the Hamiltonian are related as $\boldsymbol{v} = \partial H/\partial \boldsymbol{p}$. Under this assumption, $\mu_{\mathrm{orb}} = (e/2c)[\boldsymbol{r} \times \partial H/\partial \boldsymbol{p}]$. However, in the presence of SOC, the relation $\boldsymbol{v} = \partial H/\partial \boldsymbol{p}$, in general, does not hold. Therefore, the "modern theory of orbital magnetization" in its standard formulation cannot be used to compute relativistic contributions to orbital magnetization (when $\boldsymbol{v} \neq \partial H/\partial \boldsymbol{p}$).

At the same time, if one applies the existing nonrelativistic formulas of the "modern theory of orbital magnetization" to fully relativistic Hamiltonian and bands, this will correctly reproduce all relativistic contributions to magnetization, of both spin and orbital nature. Indeed, $\boldsymbol{\mu}^{(\mathrm{D})}$ defined in Eq. (2) is equal to $(e/2c)[\boldsymbol{r} \times \partial H^{(\mathrm{D})}/\partial \boldsymbol{p}]$ which is precisely the combination used to derive the nonrelativistic "modern theory of orbital magnetization". Moreover, it is clear that $\boldsymbol{\mu}^{(\mathrm{D})}$ does not distinguish between spin and orbital magnetic moments. Thus one can even say that, in the relativistic picture, the "modern theory of orbital magnetization" becomes the "modern theory of total magnetization". It is interesting that the expression of Eq. (2) is very rarely mentioned in the literature, we found just a few mentions [22–28].

---

[4]Here, magnetization is understood as a spatial average of magnetic moment.

## 2.5  Abnormal magnetic moment

As we have shown, the magnetic moment operator $\boldsymbol{\mu}$ is, in general, different from $-\partial H/\partial \boldsymbol{B}$ if the Hamiltonian $H$ contains relativistic corrections. This is a consequence of the fact that the operation $\partial/\partial \boldsymbol{B}$ is not invariant under unitary transformations that depend on $\boldsymbol{B}$. The velocity operator, in general, is not equal to $\partial H/\partial \boldsymbol{p}$ for the similar reason [11]. Traditionally, the difference $\boldsymbol{v} - \partial H/\partial \boldsymbol{p}$ is called the "anomalous velocity" [11, 29, 30], either in the operator form or when projected to the bands. Since the term "anomalous magnetic moment" is already used to denote the deviation of the electron g-factor from 2 [31, 32], we suggest to call the difference

$$\boldsymbol{\mu}_{\text{abn}} = \boldsymbol{\mu} - \left(-\frac{\partial H}{\partial \boldsymbol{B}}\right) \tag{17}$$

the "abnormal magnetic moment" (abn). In the Pauli picture, with the $1/c^2$ accuracy, we have

$$\boldsymbol{\mu}_{\text{abn}}^{(\text{el})} = \frac{\mu_{\text{B}}}{4mc^2}\left([\boldsymbol{r} \times [\boldsymbol{\nabla} V \times \boldsymbol{\sigma}]] - \frac{1}{m}[\boldsymbol{\pi} \times [\boldsymbol{\pi} \times \boldsymbol{\sigma}]]\right), \tag{18}$$

as it follows from Eqs. (13), (15).

## 3  Operator of magnetic moment in the Kane model

Having analyzed the electron magnetic moment in vacuum, we now turn to a more material related case. In this Section, we consider the 8 band Kane model [14] of semiconductors with the zinc blende symmetry (e.g., GaAs, InSb, InAs). Different forms and modifications of this model describe, in particular, topological insulators [33, 34] and certain Rashba materials [10, 35].

### 3.1  Diagonalization of the Kane Hamiltonian

The Kane model is formulated in the envelope function approximation [9, 36–40]. It considers one conduction and three valence bands, which is 8 bands in total when the electron spin is included. The eigenbasis of the model is spanned by the atomic functions with the spatial periodicity of the crystal. They are usually denoted as $S_\uparrow$, $S_\downarrow$, $X_\uparrow$, $X_\downarrow$, $Y_\uparrow$, $Y_\downarrow$, $Z_\uparrow$, $Z_\downarrow$, where the subscripts $\uparrow, \downarrow$ correspond to spin up and spin down states, respectively. The $S$ functions have the atomic $s$ symmetry and describe the conduction band, while the $X$, $Y$, and $Z$ functions are of the $p$ type and correspond to the valence bands.

In the standard basis that, in the absence of magnetic fields, diagonalizes the valence bands[5], the Kane (K) model Hamiltonian is given by the operator-valued block matrix [12]

$$H^{(\text{K})} = \begin{pmatrix} H_s & h \\ h^\dagger & H_{\text{g}} + H_p \end{pmatrix}, \tag{19}$$

where the Hamiltonians[6] $H_s = \hbar^2 \boldsymbol{k}^2/2m + V + \mu_{\text{B}} \boldsymbol{\sigma} \boldsymbol{B}$ and $H_p = \hbar^2 \boldsymbol{k}^2/2m + V + \mu_{\text{B}}(\boldsymbol{\Sigma} + \boldsymbol{\Lambda})\boldsymbol{B}$ act on the envelope functions in the conduction and the valence bands, respectively. Here $V$ denotes the scalar potential, of which we disregard all spatial derivatives except the first. The envelope functions' momentum operator is $\hbar \boldsymbol{k} = -i\hbar \boldsymbol{\nabla} - e\boldsymbol{A}/c$, where $\boldsymbol{A} = [\boldsymbol{B} \times \boldsymbol{r}]/2$ is the vector potential that corresponds[7] to a constant magnetic field $\boldsymbol{B}$. Vectors $\boldsymbol{\sigma}$ and $\boldsymbol{\Sigma}$ are composed of the spin operators' matrices in the respective bands, while $\hbar \boldsymbol{\Lambda}$ is the atomic orbital moment

---

[5]It is given by Eq. (15) of Ref. [12].

[6]We shifted the energy origin by a constant for all bands.

[7]One can also add a position-independent $\boldsymbol{A}_0(t)$ to the vector potential without having to modify the expression for $h$. Generalization to arbitrary dependence of $\boldsymbol{A}$ on spatial coordinates is technically straightforward

of the $p$ functions. The diagonal invertible matrix $H_g$ describes separation of the bands and $h$ quantifies the coupling between them. Expressions for $H_g$ and $h$ are provided by Eqs. (17) and (18) of Ref. [12], whereas $\Sigma$ is defined in Appendix A of the same work and $\boldsymbol{\sigma}$ is a vector of Pauli matrices. As $\Lambda$ was disregarded in Ref. [12], we give it a full consideration in Appendix A of the present paper.

Diagonalization of $H^{(K)}$ is performed as an expansion with respect to inverse powers of $H_g$. Employing a unitary transformation of the same form as in Eq. (4), one can express the transformed Hamiltonian, $H_U^{(K)} = U^\dagger H^{(K)} U$, as

$$H_U^{(K)} = \begin{pmatrix} H^{(c)} & 0 \\ 0 & H^{(v)} \end{pmatrix}, \tag{20}$$

where the (new) conduction (c) and valence (v) bands' Hamiltonians, $H^{(c)}$ and $H^{(v)}$, are decoupled from each other (with the desired accuracy). To the order $H_g^{-2}$, the unitary transformation providing this result is determined by [12]

$$w = -ihH_g^{-1} + ihH_g^{-1}H_p H_g^{-1} - iH_s h H_g^{-2}, \tag{21}$$

while the conduction band Hamiltonian reads [12]

$$H^{(c)} = \frac{\hbar^2 \boldsymbol{k}^2}{2m^*} + \lambda_{\text{SOC}}[\boldsymbol{\nabla}V \times \boldsymbol{k}]\boldsymbol{\sigma} + V + \frac{g^*}{2}\mu_B \boldsymbol{\sigma}\boldsymbol{B} + \mu_B\left[\lambda_+ \boldsymbol{k}^2(\boldsymbol{\sigma}\boldsymbol{B}) + \lambda_-(\boldsymbol{k}\boldsymbol{B})(\boldsymbol{k}\boldsymbol{\sigma})\right], \tag{22}$$

with nonlinear powers of $\boldsymbol{B}$ disregarded, including those comprised by $\boldsymbol{k}$. The expression for the effective mass $m^*$ [12, 41, 42] is not important for us here, whereas for the remaining parameters in Eq. (22) we can write [12]

$$\lambda_{\text{SOC}} = \lambda_2 + \frac{2\lambda_1}{E_g + \Delta/3}, \quad \lambda_\pm = \frac{\lambda}{2} \pm \lambda_{\text{SOC}}, \quad \frac{g^*}{2} = 1 + \frac{2m\lambda_1}{\hbar^2}, \tag{23}$$

where $E_g$ denotes the energy difference between the conduction band and the top two valence bands, $\Delta$ is the splitting between the latter and the lower valence band,

$$\lambda_n = \frac{P^2}{3}\left[\frac{1}{(E_g + \Delta)^n} - \frac{1}{E_g^n}\right], \quad \lambda = -\frac{2P^2}{9}\left[\frac{1}{E_g + \Delta} - \frac{1}{E_g}\right]^2, \tag{24}$$

and we employed the real-valued matrix elements

$$P = \frac{\hbar}{im}\langle S_{\uparrow,\downarrow}|p_x|X_{\uparrow,\downarrow}\rangle = \frac{\hbar}{im}\langle S_{\uparrow,\downarrow}|p_y|Y_{\uparrow,\downarrow}\rangle = \frac{\hbar}{im}\langle S_{\uparrow,\downarrow}|p_z|Z_{\uparrow,\downarrow}\rangle \tag{25}$$

of the atomic momentum operator $\boldsymbol{p}$. We note that the expression for $\lambda_\pm$ in Eq. (23) differs from that of Eq. (32) of Ref. [12]. This is because we disregarded the atomic orbital moment $\hbar\Lambda$ of the $p$ type functions in that work. Consistent treatment of $\Lambda$ is provided in Appendix A.

As a result of the band decoupling, the conduction band Hamiltonian acquired relativistic (or, one may say, effectively relativistic) corrections. They are represented by the 2nd and the 5th terms in Eq. (22) as well as by the modified mass and g-factor. We are now interested in computing corrections to the magnetic moment operator of electrons in the conduction band.

## 3.2 Total and abnormal magnetic moments in the conduction band

Similar to the Dirac picture, the total magnetic moment operator in the Kane model can be represented by a commutator:

$$\boldsymbol{\mu}^{(K)} = -\frac{\partial H^{(K)}}{\partial \boldsymbol{B}} = [\boldsymbol{\mathfrak{B}}^{(K)}, H^{(K)}], \quad \boldsymbol{\mathfrak{B}}^{(K)} = \begin{pmatrix} \boldsymbol{\mathfrak{B}} & 0 \\ 0 & \boldsymbol{\mathfrak{B}} \end{pmatrix}, \quad \boldsymbol{\mathfrak{B}} = -\frac{\partial}{\partial \boldsymbol{B}}. \tag{26}$$

The unitary transformed operators $\boldsymbol{\mu}_U^{(\mathrm{K})} = U^\dagger \boldsymbol{\mu}^{(\mathrm{K})} U$ and $\boldsymbol{\mathfrak{B}}_U^{(\mathrm{K})} = U^\dagger \boldsymbol{\mathfrak{B}}^{(\mathrm{K})} U$ read

$$\boldsymbol{\mu}_U^{(\mathrm{K})} = \begin{pmatrix} \boldsymbol{\mu}^{(\mathrm{c})} & \delta\boldsymbol{\mu} \\ \delta\boldsymbol{\mu}^\dagger & \boldsymbol{\mu}^{(\mathrm{v})} \end{pmatrix}, \quad \boldsymbol{\mathfrak{B}}_U^{(\mathrm{K})} = \begin{pmatrix} \boldsymbol{\mathfrak{B}}^{(\mathrm{c})} & \delta\boldsymbol{\mathfrak{B}} \\ \widetilde{\delta}\boldsymbol{\mathfrak{B}} & \boldsymbol{\mathfrak{B}}^{(\mathrm{v})} \end{pmatrix}. \tag{27}$$

Thus, since $H^{(\mathrm{c})}$ is diagonal, the electron magnetic moment operator in the conduction band can be computed by means of the relation $\boldsymbol{\mu}^{(\mathrm{c})} = [\boldsymbol{\mathfrak{B}}^{(\mathrm{c})}, H^{(\mathrm{c})}]$.

Analogous to Eq. (11), one can write, to the order $H_{\mathrm{g}}^{-2}$,

$$\boldsymbol{\mathfrak{B}}^{(\mathrm{c})} = \boldsymbol{\mathfrak{B}} - \frac{1}{2}\left([\boldsymbol{\mathfrak{B}}, w]w^\dagger - w[\boldsymbol{\mathfrak{B}}, w^\dagger]\right) = \boldsymbol{\mathfrak{B}} + \frac{1}{2}\left[\frac{\partial h}{\partial \boldsymbol{B}} H_{\mathrm{g}}^{-2} h^\dagger - h H_{\mathrm{g}}^{-2} \frac{\partial h^\dagger}{\partial \boldsymbol{B}}\right], \tag{28}$$

where we took into account Eq. (21). Using Eqs. (29) from Ref. [12] and the symmetric gauge for $\boldsymbol{B}$, we find

$$\boldsymbol{\mathfrak{B}}^{(\mathrm{c})} = \boldsymbol{\mathfrak{B}} + \frac{im\mu_{\mathrm{B}}\lambda_2}{2\hbar^2}\left([\boldsymbol{r} \times [\boldsymbol{k} \times \boldsymbol{\sigma}]] - [[\boldsymbol{k} \times \boldsymbol{\sigma}] \times \boldsymbol{r}]\right), \tag{29}$$

which is of exactly the same form as $\boldsymbol{\mathfrak{B}}^{(\mathrm{el})}$ in Eq. (12). By commuting the second term above with[8] $\hbar^2 \boldsymbol{k}^2/2m$ and $V$ in $H^{(\mathrm{c})}$, we observe that the abnormal magnetic moment operator in the conduction band of the Kane model,

$$\boldsymbol{\mu}_{\mathrm{abn}}^{(\mathrm{c})} = \frac{m\mu_{\mathrm{B}}\lambda_2}{\hbar^2}\left([\boldsymbol{r} \times [\boldsymbol{\nabla}V \times \boldsymbol{\sigma}]] - \frac{\hbar^2}{m}[\boldsymbol{k} \times [\boldsymbol{k} \times \boldsymbol{\sigma}]]\right), \tag{30}$$

up to a constant prefactor, coincides with the one in the Pauli picture, Eq. (18). For the total magnetic moment, we then obtain

$$\boldsymbol{\mu}^{(\mathrm{c})} = \boldsymbol{\mu}_{\mathrm{abn}}^{(\mathrm{c})} - \frac{\partial H^{(\mathrm{c})}}{\partial \boldsymbol{B}} = -\mu_{\mathrm{B}}\left[\frac{g^*}{2}\boldsymbol{\sigma} + \frac{1}{\hbar}\left[\boldsymbol{r} \times m\boldsymbol{v}^{(\mathrm{c})}\right] + (\lambda_+ - \lambda_2)\boldsymbol{k}^2\boldsymbol{\sigma} + (\lambda_- + \lambda_2)\boldsymbol{k}(\boldsymbol{k}\boldsymbol{\sigma})\right], \tag{31}$$

where the velocity operator

$$\boldsymbol{v}^{(\mathrm{c})} = \frac{\hbar\boldsymbol{k}}{m^*} - \frac{1}{\hbar}(\lambda_{\mathrm{SOC}} + \lambda_2)[\boldsymbol{\nabla}V \times \boldsymbol{\sigma}] \tag{32}$$

contains the anomalous velocity, i.e. differs from $\partial H^{(\mathrm{c})}/\partial(\hbar\boldsymbol{k})$ [12].

The last two terms inside the brackets in Eq. (31) may seem as small corrections to the first term there. They, however, have an important property: both couple spin and orbital degrees of freedom and thus can be interpreted as instances of SOC. In the presence of inversion asymmetry, the $\propto \boldsymbol{k}(\boldsymbol{k}\boldsymbol{\sigma})$ term can couple, e.g., to an exchange field and generate relativistic kinetic magnetoelectric effect as a result. This is different from the often considered mechanism of this effect that originates in the coupling between the spin magnetic moment $\boldsymbol{\sigma}$ (first term inside the brackets in Eq. (31)) and SOC of either Rashba [43] or Dresselhaus [44] type present in the system Hamiltonian and in the velocity operator [2, 45]. In other words, SOC-induced magnetism does not have to rely on SOC terms in the Hamiltonian and can exist without them.

## 3.3  What is orbital magnetic moment and what is spin magnetic moment?

It also instructive to learn how the spin and orbital magnetic moments in the Kane model transform separately. Let us study the former first. Prior to the unitary transformation application, the spin magnetic moment operator reads

$$\boldsymbol{\mu}_{\mathrm{spin}}^{(\mathrm{K})} = -\mu_{\mathrm{B}}\begin{pmatrix} \boldsymbol{\sigma} & 0 \\ 0 & \boldsymbol{\Sigma} \end{pmatrix}, \tag{33}$$

---

[8] $1/m - 1/m^*$ is of the order $H_{\mathrm{g}}^{-1}$ and gets disregarded here for this reason.

which can also be expressed as a derivative of the Kane Hamiltonian over the exchange field $\boldsymbol{B}_{\text{exc}}$ (with a minus sign) when the latter is taken into account. It follows from Eqs. (24), (28) of Ref. [12] that, to the order $H_{\text{g}}^{-2}$, such a derivative is interchangeable with the Hamiltonian transformation described in Sec. 3.1 because the exchange field $\boldsymbol{B}_{\text{exc}}$ appears in $w$ only as a contribution to $H_p$. Therefore, the spin magnetic moment operator of conduction band electrons (in the unitary transformed basis) is equal to $-\partial H^{(\text{c})}/\partial \boldsymbol{B}_{\text{exc}}$, where $H^{(\text{c})}$ is given by Eq. (31) of Ref. [12],

$$\boldsymbol{\mu}_{\text{spin}}^{(\text{c})} = -\mu_{\text{B}}\left(\boldsymbol{\sigma} + \lambda\left[k^2\boldsymbol{\sigma} + \boldsymbol{k}(\boldsymbol{k}\boldsymbol{\sigma})\right]\right). \tag{34}$$

By subtracting this result from Eq. (31), we also obtain the orbital magnetic moment operator in the conduction band,

$$\boldsymbol{\mu}_{\text{orb}}^{(\text{c})} = -\mu_{\text{B}}\left[\left(\frac{g^*}{2}-1\right)\boldsymbol{\sigma} + \frac{1}{\hbar}\left[\boldsymbol{r} \times m\boldsymbol{v}^{(\text{c})}\right] + (\lambda_+ - \lambda_2 - \lambda)k^2\boldsymbol{\sigma} + (\lambda_- + \lambda_2 - \lambda)\boldsymbol{k}(\boldsymbol{k}\boldsymbol{\sigma})\right], \tag{35}$$

which of course can be computed directly by transforming $\boldsymbol{\mu}_{\text{orb}}^{(\text{K})} = (-\partial H^{(\text{K})}/\partial \boldsymbol{B}) - \boldsymbol{\mu}_{\text{spin}}^{(\text{K})}$.

As one can see, the transformed orbital magnetic moment operator does not boil down to the form $\boldsymbol{r} \times \dots$. Moreover, its $\propto (g^*/2-1)\boldsymbol{\sigma}$ contribution is clearly of the "spin type", yet originates in the orbital moment $\boldsymbol{\mu}_{\text{orb}}^{(\text{K})}$. Therefore, separation of the total magnetic moment into the spin and orbital contributions is ambiguous and depends on the chosen representation[9]. Extra care should be taken when one disregards either of these contributions. Note also that $\delta g = g^*/2-1 \propto \lambda_1$ is precisely the relativistic renormalization of the g-factor in the conduction band. In semiconductors with narrow gap, it can be much larger than the vacuum g-factor [46]. Technically, $\delta g$ in Eq. (35) stems from a vector product of the interband matrix elements of the unitary transformed position and velocity operators.

## 3.4 Commutation property of spin operators

From Eqs. (33), (34), we can also see that relativistic corrections to the spin operators in the conduction band violate the spin momentum commutation relation. Indeed, while for the spin operators in the Kane model the latter is valid,[10]

$$\boldsymbol{S}^{(\text{K})} = \frac{\hbar}{2}\begin{pmatrix} \boldsymbol{\sigma} & 0 \\ 0 & \boldsymbol{\Sigma} \end{pmatrix}, \quad [S_i^{(\text{K})}, S_j^{(\text{K})}] - i\hbar\epsilon_{ijq}S_q^{(\text{K})} = 0, \tag{36}$$

for the conduction band projections, it is not,

$$[S_i^{(\text{c})}, S_j^{(\text{c})}] - i\hbar\epsilon_{ijq}S_q^{(\text{c})} = -i\lambda\hbar^2\epsilon_{ijq}[\boldsymbol{k} \times [\boldsymbol{k} \times \boldsymbol{\sigma}]]_q, \tag{37}$$

where $\epsilon_{ijq}$ denotes the Levi-Civita symbol in three dimensions and

$$\boldsymbol{S}^{(\text{c})} = \frac{\hbar}{2}\left(\boldsymbol{\sigma} + \lambda\left[k^2\boldsymbol{\sigma} + \boldsymbol{k}(\boldsymbol{k}\boldsymbol{\sigma})\right]\right) \tag{38}$$

is the spin momentum operator in the conduction band (after the band decoupling). Namely, $\boldsymbol{S}^{(\text{c})}$ is the $(1,1)$ component of $\boldsymbol{S}_U^{(\text{K})} = U^\dagger \boldsymbol{S}^{(\text{K})} U$.

It should be understood that unitary transformations preserve commutators and thus, for $\boldsymbol{S}_U^{(\text{K})}$, the spin commutation relation also holds. However, $\boldsymbol{S}_U^{(\text{K})}$, in contrast with $\boldsymbol{S}^{(\text{K})}$, is not diagonal. Commutator of the $i$ and $j$ components of its interband matrix elements coincides (up to a minus sign) with the right hand side of Eq. (37).

---

[9]It is also unclear if the terms $\propto k_i k_j \sigma_q$ in Eq. (35) should be understood as of the spin type or of the orbital type.
[10]In Eqs. (36), (37), summation over $q$ is implied.

# 4  Linear response theory for the kinetic magnetoelectric effect

So far, we have analyzed operators' projections to the bands that were decoupled from each other. As a result of the decoupling, projected operators obtained corrections, in particular in the form of SOC corrections. In the single particle picture, operating with these projections, including the projected Hamiltonian, is sufficient to study equilibrium properties of the system.

At the same time, unitary transformation that diagonalizes the Hamiltonian, also creates (or modifies) interband matrix elements of other operators. For the magnetic moment operator, for example, we denoted them as $\delta\boldsymbol{\mu}$ and $\delta\boldsymbol{\mu}^\dagger$ in Eqs. (8), (27). As it turns out, a proper time-dependent perturbation theory should take such matrix elements into account. In particular, it means that linear response formulas cannot be correctly derived in a theory that considers only the band-diagonal operators' projections (that we considered so far). In certain cases, however, the interband matrix elements in such formulas can be eventually expressed by means of the band-diagonal[11] projections. In this Section, we demonstrate how this is achieved for the the linear response of the total magnetic moment to the external time-dependent electric field. We will consider the dc response, i.e. the $\omega \to 0$ limit, but the results can be generalized to finite frequencies.

## 4.1  Model and operators

As a generalization of the two models examined in previous Sections, we consider an arbitrary electron system that can be modeled by a single particle operator-valued block matrix Hamiltonian

$$H = \begin{pmatrix} H^\Uparrow & h \\ h^\dagger & H^\Downarrow \end{pmatrix}. \tag{39}$$

We assume that there exists a unitary transformation $U$ that decouples the $\Uparrow$ and $\Downarrow$ components[12], so that the unitary transformed Hamiltonian $H_U = U^\dagger H U$ can be considered block-diagonal,

$$H_U = \begin{pmatrix} H^{(t)} & 0 \\ 0 & H^{(b)} \end{pmatrix}, \tag{40}$$

with a certain accuracy. Here, $H^{(t)}$ and $H^{(b)}$ denote the Hamiltonians of what we call the top (t) and bottom (b) branches. Each branch can represent any number of actual physical bands, not necessarily just one. For the Kane model, for example, the bottom branch would comprise all 6 valence bands. We further assume that the spectrum of $H^{(t)}$ is bounded from below by $\varepsilon^{(t)}$ and that the spectrum of $H^{(b)}$ is bounded from above by $\varepsilon^{(b)}$, with $\varepsilon^{(t)} > \varepsilon^{(b)}$.

The retarded (+) and advanced (−) Green's functions $G_{U,\pm} = (\varepsilon - H_U \pm i0)^{-1}$, as one can see from Eq. (40), are expressed by the Green's functions of the branches,

$$G_{U,\pm} = \begin{pmatrix} G_\pm^{(t)} & 0 \\ 0 & G_\pm^{(b)} \end{pmatrix}, \quad G_\pm^{(t,b)} = \left( \varepsilon - H^{(t,b)} \pm i0 \right)^{-1}. \tag{41}$$

Moreover, the retarded and advanced Green's functions of a particular branch coincide when the energy parameter lies outside of the spectrum of this branch's Hamiltonian. This fact is crucial for splitting of Kubo formulas into contributions associated with different branches. To reflect it, we introduce the following notations:

$$G_+^{(t)} = G_-^{(t)} = \left( \varepsilon - H^{(t)} \right)^{-1} \equiv G^{(t)}, \quad \text{for } \varepsilon < \varepsilon^{(t)}, \tag{42a}$$

$$G_+^{(b)} = G_-^{(b)} = \left( \varepsilon - H^{(b)} \right)^{-1} \equiv G^{(b)}, \quad \text{for } \varepsilon > \varepsilon^{(b)}. \tag{42b}$$

---

[11]Or branch-diagonal, see Sec. 4.1.
[12]Note that $\Uparrow$ and $\Downarrow$ do not refer to spin.

The total magnetic moment operator $\boldsymbol{\mu} = -\partial H/\partial \boldsymbol{B}$ is, once again, expressed as a commutator, $\boldsymbol{\mu} = [\mathfrak{B}, H]$, where $\mathfrak{B} = -\partial/\partial \boldsymbol{B}$[13]. In the unitary transformed picture, we have

$$\boldsymbol{\mu}_U = [\mathfrak{B}_U, H_U], \quad \boldsymbol{\mu}_U = \begin{pmatrix} \boldsymbol{\mu}^{(\mathrm{t})} & \delta\boldsymbol{\mu} \\ \delta\boldsymbol{\mu}^\dagger & \boldsymbol{\mu}^{(\mathrm{b})} \end{pmatrix}, \quad \mathfrak{B}_U = \begin{pmatrix} \mathfrak{B}^{(\mathrm{t})} & \delta\mathfrak{B} \\ \widetilde{\delta}\mathfrak{B} & \mathfrak{B}^{(\mathrm{b})} \end{pmatrix}. \tag{43}$$

From Eq. (40), (43), one can immediately derive how the components of $\boldsymbol{\mu}_U$ and $\mathfrak{B}_U$ are related:

$$\begin{pmatrix} \boldsymbol{\mu}^{(\mathrm{t})} & \delta\boldsymbol{\mu} \\ \delta\boldsymbol{\mu}^\dagger & \boldsymbol{\mu}^{(\mathrm{b})} \end{pmatrix} = \begin{pmatrix} \left[\mathfrak{B}^{(\mathrm{t})}, H^{(\mathrm{t})}\right] & \delta\mathfrak{B} H^{(\mathrm{b})} - H^{(\mathrm{t})}\delta\mathfrak{B} \\ \widetilde{\delta}\mathfrak{B} H^{(\mathrm{t})} - H^{(\mathrm{b})}\widetilde{\delta}\mathfrak{B} & \left[\mathfrak{B}^{(\mathrm{b})}, H^{(\mathrm{b})}\right] \end{pmatrix}. \tag{44}$$

Finally, we assume that there exists an operator $\mathfrak{r}$, such that the velocity operator can be defined as a commutator, $\boldsymbol{v} = [\mathfrak{r}, H]/(i\hbar)$. In the Dirac theory, $\mathfrak{r}$ is a physical position operator, while in the Kane model $\mathfrak{r}$ corresponds to a fictitious position [12]. Analogous to Eq. (43), we can write

$$\boldsymbol{v}_U = \frac{[\mathfrak{r}_U, H_U]}{i\hbar}, \quad \boldsymbol{v}_U = \begin{pmatrix} \boldsymbol{v}^{(\mathrm{t})} & \delta\boldsymbol{v} \\ \delta\boldsymbol{v}^\dagger & \boldsymbol{v}^{(\mathrm{b})} \end{pmatrix}, \quad \mathfrak{r}_U = \begin{pmatrix} \mathfrak{r}^{(\mathrm{t})} & \delta\mathfrak{r} \\ \delta\mathfrak{r}^\dagger & \mathfrak{r}^{(\mathrm{b})} \end{pmatrix}, \tag{45}$$

where the components of $\boldsymbol{v}_U$ and $\mathfrak{r}_U$ are related by

$$\begin{pmatrix} \boldsymbol{v}^{(\mathrm{t})} & \delta\boldsymbol{v} \\ \delta\boldsymbol{v}^\dagger & \boldsymbol{v}^{(\mathrm{b})} \end{pmatrix} = \frac{1}{i\hbar} \begin{pmatrix} \left[\mathfrak{r}^{(\mathrm{t})}, H^{(\mathrm{t})}\right] & \delta\mathfrak{r} H^{(\mathrm{b})} - H^{(\mathrm{t})}\delta\mathfrak{r} \\ \delta\mathfrak{r}^\dagger H^{(\mathrm{t})} - H^{(\mathrm{b})}\delta\mathfrak{r}^\dagger & \left[\mathfrak{r}^{(\mathrm{b})}, H^{(\mathrm{b})}\right] \end{pmatrix}. \tag{46}$$

We additionally require the Cartesian coordinates of $\mathfrak{B}$ and $\mathfrak{r}$ to commute, i.e. $[\mathfrak{B}_\zeta, \mathfrak{r}_\eta] = 0$.[14]

## 4.2 Kubo formula "projected" to the branches

We will now analyze the linear response Kubo formula for the kinetic magnetoelectric effect [15, 16]. The latter represents the situation when an external time-dependent electric field $\boldsymbol{E}$ induces magnetization $\boldsymbol{M}$ in the system. In a linear theory, this effect is described by a magnetoelectric tensor $\chi_{\zeta\eta}$ such that $M_\zeta = \chi_{\zeta\eta} E_\eta$. Using the $\boldsymbol{E} = -(1/c)\partial \boldsymbol{A}/\partial t$ gauge and assuming $[\boldsymbol{v}, \boldsymbol{E}] = 0$, we can derive a Kubo formula for the dc magnetoelectric tensor,

$$\chi_{\zeta\eta} = \frac{e\hbar}{2\pi\Omega} \mathrm{Tr} \int d\varepsilon\, f_\varepsilon \left[G_{U,-} - G_{U,+}\right] \mu_{U,\zeta} \left[G_{U,+}\right]' v_{U,\eta} + \mathrm{c.c.}, \tag{47}$$

where $f_\varepsilon$ is the Fermi-Dirac distribution, $\Omega$ denotes the system volume, and prime represents a derivative with respect to $\varepsilon$. Note that Eq. (47) expresses magnetization averaged over the entire system. For more spatial resolution, one can use definitions of the magnetic moment operator that correspond to magnetic fields with a certain degree of localization [47].

To "project" the Kubo formula of Eq. (47) to the branches, we expand the matrix products and compute the matrix trace. This leads to

$$\chi_{\zeta\eta} = \frac{e\hbar}{2\pi\Omega} \mathrm{tr} \int_{\varepsilon \geq \varepsilon^{(\mathrm{t})}} d\varepsilon\, f_\varepsilon \left( \left[G_-^{(\mathrm{t})} - G_+^{(\mathrm{t})}\right] \mu_\zeta^{(\mathrm{t})} \left[G_+^{(\mathrm{t})}\right]' v_\eta^{(\mathrm{t})} + \left[G_-^{(\mathrm{t})} - G_+^{(\mathrm{t})}\right] \delta\mu_\zeta \left[G^{(\mathrm{b})}\right]' \delta v_\eta^\dagger \right)$$

$$+ \frac{e\hbar}{2\pi\Omega} \mathrm{tr} \int_{\varepsilon \leq \varepsilon^{(\mathrm{b})}} d\varepsilon\, f_\varepsilon \left( \left[G_-^{(\mathrm{b})} - G_+^{(\mathrm{b})}\right] \delta\mu_\zeta^\dagger \left[G^{(\mathrm{t})}\right]' \delta v_\eta + \left[G_-^{(\mathrm{b})} - G_+^{(\mathrm{b})}\right] \mu_\zeta^{(\mathrm{b})} \left[G_+^{(\mathrm{b})}\right]' v_\eta^{(\mathrm{b})} \right) + \mathrm{c.c.}, \tag{48}$$

where tr represents the operator traces computed within the branches, and we modified the integration domains using that $G_-^{(\mathrm{t})} - G_+^{(\mathrm{t})} = 0$ for $\varepsilon < \varepsilon^{(\mathrm{t})}$ and $G_-^{(\mathrm{b})} - G_+^{(\mathrm{b})} = 0$ for $\varepsilon > \varepsilon^{(\mathrm{b})}$. Eq. (48)

---

[13]Times the identity matrix.

[14]Which they do, both in the Dirac theory and in the Kane model.

contains four contributions to $\chi_{\zeta\eta}$, two of them are expressed by operators acting within the same branch, the other two involve Green's functions of both branches as well as components of the "interbranch" operators $\delta\nu$, $\delta\mu$. We would like to eliminate the latter.

This can be achieved by employing the relations

$$G_\pm^{(t)}\,\delta\mu\,G^{(b)} = \delta\mathfrak{B}G^{(b)} - G_\pm^{(t)}\delta\mathfrak{B}, \quad i\hbar G^{(b)}\delta\nu^\dagger G_\pm^{(t)} = \delta\mathfrak{r}^\dagger G_\pm^{(t)} - G^{(b)}\delta\mathfrak{r}^\dagger, \quad \text{for } \varepsilon > \varepsilon^{(b)}, \quad (49a)$$

$$G_\pm^{(b)}\delta\mu^\dagger G^{(t)} = \widetilde{\delta}\mathfrak{B}G^{(t)} - G_\pm^{(b)}\widetilde{\delta}\mathfrak{B}, \quad i\hbar G^{(t)}\delta\nu\,G_\pm^{(b)} = \delta\mathfrak{r}\,G_\pm^{(b)} - G^{(t)}\delta\mathfrak{r}, \quad \text{for } \varepsilon < \varepsilon^{(t)} \quad (49b)$$

derived from Eqs. (42), (44), and (46). Using the property $[G^{(t,b)}]' = -[G^{(t,b)}]^2$ and Eqs. (49) together with their conjugate, we find

$$\mathrm{tr}\left(\left[G_-^{(t)} - G_+^{(t)}\right]\delta\mu_\zeta\left[G^{(b)}\right]'\delta\nu_\eta^\dagger\right) + \text{c.c.} = \frac{1}{i\hbar}\mathrm{tr}\left(\left[G_-^{(t)} - G_+^{(t)}\right]\left[\delta\mathfrak{B}_\zeta\delta\mathfrak{r}_\eta^\dagger - \delta\mathfrak{r}_\eta\widetilde{\delta}\mathfrak{B}_\zeta\right]\right), \quad (50a)$$

$$\mathrm{tr}\left(\left[G_-^{(b)} - G_+^{(b)}\right]\delta\mu_\zeta^\dagger\left[G^{(t)}\right]'\delta\nu_\eta\right) + \text{c.c.} = \frac{1}{i\hbar}\mathrm{tr}\left(\left[G_-^{(b)} - G_+^{(b)}\right]\left[\widetilde{\delta}\mathfrak{B}_\zeta\delta\mathfrak{r}_\eta - \delta\mathfrak{r}_\eta^\dagger\delta\mathfrak{B}_\zeta\right]\right). \quad (50b)$$

These formulas simplify further due to the relation $[\mathfrak{B}_\zeta, \mathfrak{r}_\eta] = [\mathfrak{B}_{U,\zeta}, \mathfrak{r}_{U,\eta}] = 0$ that translates into

$$\delta\mathfrak{B}_\zeta\delta\mathfrak{r}_\eta^\dagger - \delta\mathfrak{r}_\eta\widetilde{\delta}\mathfrak{B}_\zeta = -[\mathfrak{B}_\zeta^{(t)}, \mathfrak{r}_\eta^{(t)}], \quad \widetilde{\delta}\mathfrak{B}_\zeta\delta\mathfrak{r}_\eta - \delta\mathfrak{r}_\eta^\dagger\delta\mathfrak{B}_k = -[\mathfrak{B}_\zeta^{(b)}, \mathfrak{r}_\eta^{(b)}], \quad (51)$$

leaving us with only the branch-diagonal (or intrabrach) components of all the involved operators.

By combining Eqs. (48), (50), and (51), we are able to express the dc magnetoelectric tensor as a sum of distinct contributions from the two branches, $\chi_{\zeta\eta} = \chi_{\zeta\eta}^{(t)} + \chi_{\zeta\eta}^{(b)}$, with

$$\chi_{\zeta\eta}^{(t,b)} = \chi_{\zeta\eta}^{(t,b)\text{-I}} + \chi_{\zeta\eta}^{(t,b)\text{-II}} + \chi_{\zeta\eta}^{(t,b)\text{-III}}, \quad (52a)$$

$$\chi_{\zeta\eta}^{(t,b)\text{-I}} + \chi_{\zeta\eta}^{(t,b)\text{-II}} = \frac{e\hbar}{2\pi\Omega}\mathrm{tr}\int d\varepsilon\, f_\varepsilon\left[G_-^{(t,b)} - G_+^{(t,b)}\right]\mu_\zeta^{(t,b)}\left[G_+^{(t,b)}\right]'\nu_\eta^{(t,b)}, \quad (52b)$$

$$\chi_{\zeta\eta}^{(t,b)\text{-III}} = \frac{ie}{2\pi\Omega}\mathrm{tr}\int d\varepsilon\, f_\varepsilon\left[G_-^{(t,b)} - G_+^{(t,b)}\right][\mathfrak{B}_\zeta^{(t,b)}, \mathfrak{r}_\eta^{(t,b)}], \quad (52c)$$

where the integration domains are $\varepsilon \geq \varepsilon^{(t)}$ for the top branch and $\varepsilon \leq \varepsilon^{(b)}$ for the bottom one[15]. The $\chi_{\zeta\eta}^{(t,b)\text{-I}}$ and $\chi_{\zeta\eta}^{(t,b)\text{-II}}$ tensors are analogs of the contributions to the conductivity tensor of the similar structure [11, 48],

$$\chi_{\zeta\eta}^{(t,b)\text{-I}} = \frac{e\hbar}{4\pi\Omega}\mathrm{tr}\int d\varepsilon\, f_\varepsilon'\left[G_-^{(t,b)} - G_+^{(t,b)}\right]\mu_\zeta^{(t,b)}G_+^{(t,b)}\nu_\eta^{(t,b)} + \text{c.c.}, \quad (53a)$$

$$\chi_{\zeta\eta}^{(t,b)\text{-II}} = \frac{e\hbar}{4\pi\Omega}\mathrm{tr}\int d\varepsilon\, f_\varepsilon\left[G_-^{(t,b)}\mu_\zeta^{(t,b)}G_-^{(t,b)}\nu_\eta^{(t,b)}G_-^{(t,b)} - G_-^{(t,b)}\nu_\eta^{(t,b)}G_-^{(t,b)}\mu_\zeta^{(t,b)}G_-^{(t,b)}\right] + \text{c.c.}. \quad (53b)$$

These expressions are derived from a partial integration applied to Eq. (52b). Depending on a problem, $\chi_{\zeta\eta}^{(t,b)\text{-I}}$ and $\chi_{\zeta\eta}^{(t,b)\text{-II}}$ can be analyzed either separately, using Eqs. (53), or as a sum of Eq. (52b).

If one attempted to derive a linear response Kubo formula for $\chi_{\zeta\eta}$, starting from an effective model projected to a particular branch, (t) or (b), the result would be $\chi_{\zeta\eta}^{(t,b)\text{-I}} + \chi_{\zeta\eta}^{(t,b)\text{-II}}$, while $\chi_{\zeta\eta}^{(t,b)\text{-III}}$ would be lost. It is thus necessary to take into account the full matrix structure of both the observable and the perturbation determined by the operators $\mu$ and $\nu$, respectively. Indeed, as we have shown, their interbranch elements provide contributions to the result. Fortunately, we were able to express the latter by means of only the intrabranch elements of $\mu$ and $\nu$. This is possible because both $\mu$ and $\nu$ can be represented as commutators with the Hamiltonian.

---

[15]In principal, since we have already eliminated $G^{(t)}$ and $G^{(b)}$ from $\chi_{\zeta\eta}$, the integration domains can be extended to the entire real line.

## 4.3  Magnetoelectric effect from the noncommuting $\partial/\partial B$ and position operators

The result of Eq. (52c) suggests that there exists a particular intrinsic contribution to the kinetic magnetoelectric effect given by the commutators of the intrabranch elements of $\mathfrak{B}$ and $\mathfrak{r}$ (i.e., by their Cartesian components) integrated over the spectra of the corresponding branches[16]. Let us analyze this contribution for the two cases that we considered in the previous Sections, namely for the electronic branch of the Pauli picture and for the conduction band of the Kane model. We will use the superscript (t) to relate to these particular branches. The contribution to $M$ that we are interested in is $M^{\text{(t)-III}} = e_\zeta \chi^{\text{(t)-III}}_{\zeta\eta} E_\eta$ with summation over $\zeta$ and $\eta$ implied.

In both cases under consideration, $\mathfrak{B}^{(t)}$ and $\mathfrak{r}^{(t)}$ can be expressed as (see Eqs. (12), (29) of this work, Eq. (16a) of Ref. [11], and Eq. (41) of Ref. [12])

$$\mathfrak{B}^{(t)} = -\frac{\partial}{\partial B} + \frac{im\mu_B \widetilde{\lambda}}{2\hbar^2}\big([r \times [k \times \sigma]] - [[k \times \sigma] \times r]\big), \quad \mathfrak{r}^{(t)} = r + \widetilde{\lambda}[k \times \sigma], \quad (54)$$

where $\widetilde{\lambda} = \lambda_2$ for the Kane model and $\widetilde{\lambda} = \hbar^2/4m^2c^2$ for the Pauli picture (we consider only the leading order SOC corrections). Using Eq. (54) and the symmetric gauge for $B$, we compute the commutator in Eq. (52c),

$$[\mathfrak{B}^{(t)}_\zeta, \mathfrak{r}^{(t)}_\eta] = \frac{2m\mu_B \widetilde{\lambda}}{\hbar^2}\big[r \times [e_\eta \times \sigma]\big] \cdot e_\zeta, \quad (55)$$

which, in turn, gives for the induced magnetization

$$M^{\text{(t)-III}} = -\frac{e^2}{\hbar c}\widetilde{\lambda}\,\langle[r \times [\sigma \times E]]\rangle_{\Omega,\varepsilon}, \quad \langle Q \rangle_{\Omega,\varepsilon} = \frac{1}{\Omega}\,\text{tr} \int d\varepsilon\, f_\varepsilon \frac{G^{(t)}_- - G^{(t)}_+}{2\pi i} Q. \quad (56)$$

Here, we introduced the notation $\langle \cdot \rangle_{\Omega,\varepsilon}$ for averaging over both the system volume and the occupied states (of the (t) branch).

The contribution to magnetization provided by Eq. (56) is of the same nature as the Hall current determined by the conductivity $\sigma^{\text{III}}$ tensor introduced in Ref. [11]. For the electronic branch of the Pauli picture and for the conduction band of the Kane model, this current can be computed from Eq. (54) of the present work and Eq. (24c) of Ref. [11], leading to[17]

$$j^{\text{(t)-III}} = e_\zeta \sigma^{\text{(t)-III}}_{\zeta\eta} E_\eta = -\frac{2e^2}{\hbar}\widetilde{\lambda}\,\langle[\sigma \times E]\rangle_{\Omega,\varepsilon}, \quad (57)$$

so that the $\varepsilon$-resolved contributions to $M^{\text{(t)-III}}$ and $j^{\text{(t)-III}}$ are related to each other by a conventional formula of classical electrodynamics, $M = [r \times j]/2c$. We thus see that $M^{\text{(t)-III}}$ is an orbital-like magnetic moment of a particular Hall current which is determined by the electron spin. It is not clear whether it should fall into the orbital magnetic moment category or into the spin one. But it clearly cannot be ignored if the leading order relativistic corrections are to be taken into account.

We note that the expression for $M^{\text{(t)-III}}$ obtained in this Section is model-dependent. It is the general formula of Eq. (52c) that should be considered outside of the Kane model or the Pauli picture. We also note that $M^{\text{(t)-III}}$ of Eq. (56) requires the inversion and time-reversal symmetry breaking.

---

[16]This is an analog of the contribution to the Hall conductivity from noncommuting components of the position operator, see also Ref. [11] and Eq. (57).

[17]$j^{\text{(t)-III}}$ in Eq. (57) is a Hall current proportional to $E \times M_\sigma$, where $M_\sigma$ is the spin magnetization. Such currents were theoretically studied for the Kane model [49] and measured in InSb [50]. Eq. (102) of Ref. [51] generalizes Eq. (57) to finite frequencies in the Pauli picture.

# 5    Conclusion

We presented a consistent approach for computing relativistic corrections to the operator of magnetic moment of electron. We did this for the Pauli picture, for the Kane model, and for a general "two-branch" system. We introduced the notion of relativistic corrections to the operation $\partial/\partial \boldsymbol{B}$, where $\boldsymbol{B}$ is the external magnetic field. Commutation of these corrections with the system Hamiltonian $H$ quantifies the deviation of the total magnetic moment of the system from $-\partial H/\partial \boldsymbol{B}$. We suggest to call this deviation the abnormal magnetic moment.

Both the abnormal magnetic moment and the naive one, $-\partial H/\partial \boldsymbol{B}$, contain relativistic terms, in particular in the form of SOC corrections. They should be taken into account in numerical and theoretical analysis of relativistic effects involving magnetic moments of electrons, e.g. the kinetic magnetoelectric effect. While often the SOC terms of the Hamiltonian capture the main attention, the SOC terms of the observables are no less important.

For the Kane model, we obtained the explicit expressions for the spin magnetic moment and orbital magnetic moment operators projected to the conduction band, in the presence of the leading order and the subleading order relativistic corrections. Due to the corrections, the classification "spin moment — orbital moment" becomes ambiguous and basis-dependent. Relativistic corrections also violate the momentum commutation properties in the bands (or branches).

We derived the linear response Kubo formula for the dc kinetic magnetoelectric effect and "projected" this formula to the individual branches of a two branch system. We identified an intrinsic contribution to the non-equilibrium magnetization induced by the external electric field that stems from the interbranch (or interband) matrix elements of the magnetic moment and the velocity operators. It can also be expressed with the help of a commutator of the position and $\partial/\partial \boldsymbol{B}$ operators projected to the branches. For the Pauli picture and for the Kane model, this contribution corresponds to a Hall current proportional to $\boldsymbol{E} \times \boldsymbol{M}_\sigma$, where $\boldsymbol{M}_\sigma$ is the spin magnetization of the system. Note that this current is different from the so-called "magnetization current" $\propto \nabla \times \boldsymbol{M}_\sigma$.

We also computed contributions to the conduction band operators of the diagonalized Kane model from the atomic orbital moment of the $p$ functions, as this was not done in our previous work, Ref. [12].

The methodology we developed in this paper provides a general path towards consistent treatment of relativistic magnetic phenomena.

# Acknowledgements

We are grateful to Vladimir Bashmakov, Sergii Grytsiuk, Yuriy Mokrousov, Misha Katsnelson, and Achille Mauri for informative discussions.

**Funding information**    R. A. D. has received funding from the European Research Council (ERC) under the European Union's Horizon 2020 research and innovation programme (Grant No. 725509). The Research Council of Norway (RCN) supported A. B. through its Centres of Excellence funding scheme, project number 262633, "QuSpin". M. T. has received funding from the European Union's Horizon 2020 research and innovation program under the Marie Skłodowska-Curie grant agreement No 873028.

# A    Atomic orbital moment of the $p$ functions in the Kane model

The valence band Hamiltonian $H_p$ in Eq. (19) depends on the orbital moment of the atomic functions $X$, $Y$, $Z$. The latter is determined by the matrix $\boldsymbol{\Lambda}$ which is the representation of the operator

$[\boldsymbol{\rho} \times \boldsymbol{p}]/\hbar$ in the valence bands' basis given by the last 6 functions in Eq. (15) of Ref. [12]. Here $\boldsymbol{\rho}$ and $\boldsymbol{p}$ are the position and momentum operators acting solely on the atomic functions. The conduction band Hamiltonian $H_s$ does not contain terms with the atomic orbital moment because of the spatial symmetry of the $S$ function.

To compute $\boldsymbol{\Lambda}$, we first compute the matrix elements of $[\boldsymbol{\rho} \times \boldsymbol{p}]$ between the $p$ functions. For this, we use that $(X + iY)/\sqrt{2}$ is an eigenfunction of the $z$ component of the orbital moment operator with the eigenvalue equal to $\hbar$. Therefore,

$$\hbar = \left\langle \frac{X+iY}{\sqrt{2}} \middle| \rho_x p_y - \rho_y p_x \middle| \frac{X+iY}{\sqrt{2}} \right\rangle = i \left\langle X | \rho_x p_y - \rho_y p_x | Y \right\rangle, \tag{58}$$

where we used that $X$ and $Y$ are both real-valued and that $\langle X|[\boldsymbol{\rho} \times \boldsymbol{p}]|X\rangle = \langle Y|[\boldsymbol{\rho} \times \boldsymbol{p}]|Y\rangle = 0$. Extending this result by symmetry to the $Z$ function and also to other components of $[\boldsymbol{\rho} \times \boldsymbol{p}]$, we find that, in the basis $\{X, Y, Z\}$, the matrix of the operator $[\boldsymbol{\rho} \times \boldsymbol{p}]$ is represented by the elements $-i\hbar\epsilon_{ijk}\boldsymbol{e}_k$. Switching to the basis given by the last 6 functions in Eq. (15) of Ref. [12], we obtain

$$\Lambda_x = \begin{pmatrix} 0 & 0 & -\frac{1}{\sqrt{3}} & 0 & 0 & -\frac{1}{\sqrt{6}} \\ 0 & 0 & 0 & \frac{1}{\sqrt{3}} & \frac{1}{\sqrt{6}} & 0 \\ -\frac{1}{\sqrt{3}} & 0 & 0 & \frac{2}{3} & -\frac{1}{3\sqrt{2}} & 0 \\ 0 & \frac{1}{\sqrt{3}} & \frac{2}{3} & 0 & 0 & -\frac{1}{3\sqrt{2}} \\ 0 & \frac{1}{\sqrt{6}} & -\frac{1}{3\sqrt{2}} & 0 & 0 & -\frac{2}{3} \\ -\frac{1}{\sqrt{6}} & 0 & 0 & -\frac{1}{3\sqrt{2}} & -\frac{2}{3} & 0 \end{pmatrix}, \quad \Lambda_y = \begin{pmatrix} 0 & 0 & \frac{i}{\sqrt{3}} & 0 & 0 & \frac{i}{\sqrt{6}} \\ 0 & 0 & 0 & \frac{i}{\sqrt{3}} & \frac{i}{\sqrt{6}} & 0 \\ -\frac{i}{\sqrt{3}} & 0 & 0 & -\frac{2i}{3} & \frac{i}{3\sqrt{2}} & 0 \\ 0 & -\frac{i}{\sqrt{3}} & \frac{2i}{3} & 0 & 0 & -\frac{i}{3\sqrt{2}} \\ 0 & -\frac{i}{\sqrt{6}} & -\frac{i}{3\sqrt{2}} & 0 & 0 & -\frac{2i}{3} \\ -\frac{i}{\sqrt{6}} & 0 & 0 & \frac{i}{3\sqrt{2}} & \frac{2i}{3} & 0 \end{pmatrix},$$

$$\Lambda_z = \begin{pmatrix} 1 & 0 & 0 & 0 & 0 & 0 \\ 0 & -1 & 0 & 0 & 0 & 0 \\ 0 & 0 & \frac{1}{3} & 0 & 0 & -\frac{\sqrt{2}}{3} \\ 0 & 0 & 0 & -\frac{1}{3} & \frac{\sqrt{2}}{3} & 0 \\ 0 & 0 & 0 & \frac{\sqrt{2}}{3} & -\frac{2}{3} & 0 \\ 0 & 0 & -\frac{\sqrt{2}}{3} & 0 & 0 & \frac{2}{3} \end{pmatrix}.$$

Note that, in this basis, the (dimensionless) $z$ component of the total moment is a diagonal matrix,

$$J_z = \Lambda_z + \frac{1}{2}\Sigma_z = \mathrm{diag}\left(\frac{3}{2}, -\frac{3}{2}, \frac{1}{2}, -\frac{1}{2}, -\frac{1}{2}, \frac{1}{2}\right). \tag{59}$$

We now want to derive the contribution from the term $\mu_{\mathrm{B}}\boldsymbol{\Lambda}\boldsymbol{B}$ in $H_p$ to the conduction band Hamiltonian $H^{(\mathrm{c})}$ of the Kane model with the decoupled bands. Let us denote this contribution by $\delta_{\boldsymbol{\Lambda}}H^{(\mathrm{c})}$. We substitute the expressions for the components of $\boldsymbol{\Lambda}$ into Eq. (26) of Ref. [12] and obtain

$$\delta_{\boldsymbol{\Lambda}}H^{(\mathrm{c})} = -\mu_{\mathrm{B}}\left[\left(\frac{\lambda}{2} + \lambda_2\right)k^2(\boldsymbol{\sigma}\boldsymbol{B}) + \left(\frac{\lambda}{2} - \lambda_2\right)(\boldsymbol{k}\boldsymbol{B})(\boldsymbol{k}\boldsymbol{\sigma})\right]. \tag{60}$$

Combining Eq. (60) with Eqs. (31), (32) of Ref. [12], we arrive at Eqs. (22), (23).

We note that Eq. (C.1) of Ref. [12], which expresses the velocity operator in the conduction band, should also be updated in the presence of $\mu_{\mathrm{B}}\boldsymbol{\Lambda}\boldsymbol{B}$. Namely, for the parameters $\lambda_{\pm}$ one should be using the definitions of Eq. (23) instead of those presented in Ref. [12].

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
