# Peer review of "Magnetic moment of electrons in systems with spin-orbit coupling"

_SciPost Physics_

## Round 2 · Referee Report · Anonymous (Referee 1) · 2025-5-9

Strengths

The manuscript addresses an interesting point, the relativistic corrections to the magnetic moment of electrons. In a large number of works in the past 20 years these relativistic corrections have been ignored. The authors point out this for the kinetic magnetoelectric effect (or Edelstein effect) and the modern theory of orbital magnetization. It could thus be that additional relativistic contributions need to be considered. It is however not known how large these could be.
The authors arrive at several useful results, such as their expression for the abnormal magnetic moment . They point out the difficulty to separate in fact spin and orbital contributions, and they provide an analysis of the Kane 8 band model. Altogether, it will be interesting to see how large these relativistic contributions are in future studies of real materials.

Weaknesses

One of the weaknesses of the manuscript is that there is a body of prior work related to the "proper" definition of the relativistic spin operator , the authors are possibly not aware of it (see their statement in 4th paragraph on p. 2). For a recent summary of previous achievements, see Mondal and Oppeneer, JPCM 32, 455802 (2020). Already Foldy and Wouthuysen and Pryce formulated relativistic forms of the spin operator; these spin operators are not identical but obey all the conditions for spin operators. The relativistic total angular momentum can then be shown to be conserved for a free Dirac particle. The corresponding orbital momentum operators are then also not identical, but the sum of spin and orbital momentum is zero.

The derivation of the authors, to go from the Dirac 4-vector description to the 2-component Pauli Hamiltonian in the presence of the electro-magnetic field is in principle good. It has been done previously by several authors, see for example Mondal et al, PRB 94, 144419 (2016). Are the results of the authors consistent with other previous derivations?

The derivation of the derived Hamiltonian with respect to the B field is used as definition of the electron magnetic moment. This can be done in this way, but then one has the total magnetic moment (spin + orbital parts). I am uncertain about the procedure to take the derivative with respect to B - the quantity $\pi$ that remains after the derivative still contains $A=[B \times r]/2$ so $B$ is still in the expression? This makes the defined "abnormal magnetic moment" (Eq. (18)) dependent on the external B field.

In the Kane model, the quantity $\lambda_2$ adopts the role of a spin-orbit parameter ($\sim 1/c^2$). Is $\lambda_2$ really small in this model? I didn't see a remark on this. In the non-relativistic limit $\lambda_2$ should be zero.

There is a procedure mentioned that could be dangerous. To define the spin magnetic moment by the derivative of the Hamiltonian with respect to the exchange field. This is correct in the non relativistic limit, but in the relativistic limit there are relativistic corrections to the exchange field, see e.g. Mondal et al, PRB 94, 144419 (2016); these are practically always ignored. The authors make a derivation without exchange field but one would need to consider it already in the Dirac Hamiltonian, to find the corresponding spin moment contribution later on, when takes the derivative.

Report

The manuscript touches on topic of current interest (magnetoelectric effect, modern theory of orbital magnetism) and could be worthwhile for others working in these areas. However, the manuscript needs to be improved, see recommended changes below.

Requested changes

The authors are requested to consider previous work related to the definition of the relativistic spin operator and include this in their manuscript.
Address if their derivation provides the same result as previous derivations.
Check if their definition has become independent of B, or is the spin moment operator still dependent on B?
Check what happens with the derivation with respect to the exchange field, if there are not additional relativistic corrections to the exchange field that also might play a role. Possibly one can shift these corrections into the spin operator and then only have use the non relativistic exchange field.

Recommendation

Ask for minor revision

  • validity: good
  • significance: good
  • originality: good
  • clarity: good
  • formatting: excellent
  • grammar: excellent

Author:  Ivan Ado  on 2025-06-16  [id 5572]

(in reply to Report 1 on 2025-05-09)

We thank the Referee for reviewing our manuscript. Below we answer their questions and reply to the expressed critique.

  1. Our work does not aim to define a relativistic spin operator, relativistic spin angular momentum operator, or relativistic orbital angular momentum operator. Instead, we focus on the magnetic moment operator, which is uniquely defined as $-\partial H/\partial B$, where $B$ is the external magnetic field. In the nonrelativistic limit, this operator is conventionally expressed as a sum of spin and orbital contributions. We compute relativistic corrections to both of these terms. In a revised version of the manuscript, we will include a brief discussion clarifying the difference between our work and that of JPCM 32, 455802 (2020).

  2. The two-component Pauli Hamiltonian we derive (in the absence of an exchange field) is indeed well-established and consistent with existing literature. Our derivation reaffirms this known result.

  3. We agree with the Referee that the magnetic moment operator generally depends on B. This is analogous to how the electric current operator depends on the vector potential through the diamagnetic contribution. We will add a clarifying remark to this effect in the manuscript.

  4. In GaAs, $\lambda_2$ is approximately 6 orders of magnitude larger than the vacuum spin-orbit coupling (SOC) parameter. In InSb, the enhancement is about 7 orders of magnitude. More broadly, SOC effects in narrow-gap semiconductors are substantially stronger than in vacuum due to interband coupling. This key difference underpins the dominant role of SOC in these materials.

  5. We agree that relativistic corrections to the exchange field arise when one takes derivatives with respect to it. However, in narrow-gap semiconductors, vacuum corrections, except for the SOC-induced splitting in the valence bands (represented by Delta in Eq. (24)), are typically negligible. This is because they are several orders of magnitude smaller than corrections arising from conduction–valence band coupling. We will clarify this point and include a relevant comment in the revised manuscript.

---

## Round 2 · Referee Report · Anonymous (Referee 2) · 2025-6-14

Strengths

  1. A fully microscopic definition of the magnetic moment given
  2. Possible generalization of the existing results are discussed
  3. Examples of calculations are given

Weaknesses

  1. No direct comparison to the existing results is presented.
  2. No specific errors or omissions in the existing literature are pointed out.

Report

The work by Ado et al addresses one of the more convoluted issues in the semiclassical theory of electron motion in solid - the one of the orbital magnetic moment. It both addresses the microscopic definition of this quantity, and gives an example of a linear response calculation that involves it. I think that an attempt at "cleaning up" the situation in the literature is commendable, and should be seriously considered for publication.

Surprisingly, the paper does not include a careful comparison with the existing literature. For the first part of the paper, where the definition of the magnetic moment is considered, it would desirable to go over, say, the expressions given in a well-known review by Di Xiao and Qian Niu, Section IX of it, and compare the results of the present work to those given the review. In particular, the authors should comment on Section IX-F and IX-G. Given the extensive literature on the subject, one cannot simply add to it, there must be a critical comparison of the new results to the old ones.

Regarding the issue of the "spin" and "orbital" magnetic moment, one should point out that there is really no "spin" magnetic moment, it is all orbital, which has been elucidated by Huang in "On the Zitterbewegung of the Dirac Electron", Am. J. Phys. 20, 479–484 (1952).

Finally, it was unclear to me what the actual results of Section 4 of the paper were. The authors implied that the results of Refs 15 and 16 were incomplete, but never pointed out the errors or omissions in those works. In particular, Ref. 16 is close in spirit to the present work, so it should be critically evaluated.

I think the authors emphasized that single-band theories of KME based on the orbital magnetic moment of electrons in that particular band were incomplete. Unfortunately, I do not quite see why that is the case. The reason is the results of those one-band theories can be derived from the multi-band quantum kinetic equation, in which no reference to the magnetic moment is made at all. One just evaluates the interband coherences created by external fields, and evaluates the full microscopic current, which involves the interband matrix elements of the velocity operator. It so happens that the result can be expressed in a band-diagonal form, which involves the intrinsic orbital magnetic moment. If there are flaws in the outlined procedure, it would be great to know it - this will lead to a great deal of existing results being revised. But such serious claims need serious support.

Requested changes

I think the paper can be extremely useful for workers in the field if it makes detailed comparison of the obtained results, which often look quite convoluted, to the existing ones, with a critical analysis of errors made in the established literature. Without such an analysis the work will be another obscure contribution to a field full of misconceptions.

Recommendation

Ask for major revision

  • validity: ok
  • significance: good
  • originality: good
  • clarity: ok
  • formatting: excellent
  • grammar: excellent

Author:  Ivan Ado  on 2025-12-18  [id 6155]

(in reply to Report 2 on 2025-06-14)

We thank the Referee for reviewing our manuscript and for encouraging us to clarify how our results relate to prior work. Below we reply to the expressed critique and comment on the raised issues.

*1* We agree that more comments should have been made regarding the relation between our results and the prior ones. We made the following changes.

a) In the first part of the revised manuscript, we stressed that our definition of magnetic moment agrees with the standard one in the Dirac theory and that, in equilibrium, the magnetic moment in the Dirac theory can be computed using the “modern theory of orbital magnetization” (with relativistic bands).
b) We added a new Section in which we discuss how the Berry curvature approach relates to ours.
c) We also added a Section with a discussion of the magnetoelectric tensor computation from the current-current correlator. In short, some of the results obtained using that method coincide with ours, others seemingly do not.

*2* We understand Section IX-F of the review by Xiao, Chang, and Niu as emphasizing that a transformation of the Hamiltonian alone is generally insufficient to obtain a fully consistent effective theory. We fully agree with this point. The block-summation hierarchy scheme discussed in Section IX-G of the same review concerns traces over blocks of the matrix-valued Berry connection; while conceptually related at the level of multiband structure, it is not directly tied to the central claims of our work.

*3* We thank the Referee for pointing out Am. J. Phys. 20, 479–484 (1952). This reference uses the same Dirac-theory expression for the magnetic moment as we do, and we have added it to the bibliography. Regarding the interpretation of spin as “orbital”, our reading is different: the paper shows that for solutions of the Dirac equation one can rewrite the spin as the “orbital moment of the difference between instantaneous velocity and average velocity”. While this is an interesting observation, we do not view it as eliminating the physical distinction between the spin and orbital magnetic moments (at least in regimes where relativistic corrections do not strongly mix them). To avoid overstatement, we have shortened the “spin versus orbital” discussion in the revised manuscript.

*4* It seems to us that existing approaches for computing magnetization do not take into account the difference between the velocity operator and $\partial H/\partial\boldsymbol p$ as well as the difference between the magnetic moment operator and $-\partial H/\partial\boldsymbol B$. In the new version of the manuscript, we tried to emphasize this point.

---

## Editorial Decision

resubmitted